# Advanced Glycation End Products and Inflammation in Type 1 Diabetes Development

**DOI:** 10.3390/cells11213503

**Published:** 2022-11-04

**Authors:** Chenping Du, Rani O. Whiddett, Irina Buckle, Chen Chen, Josephine M. Forbes, Amelia K. Fotheringham

**Affiliations:** 1Glycation and Diabetes Complications Group, Mater Research Institute-The University of Queensland, Translational Research Institute, Woolloongabba 4102, Australia; 2School of Biomedical Sciences, Faculty of Medicine, The University of Queensland, St Lucia 4072, Australia; 3Faculty of Medicine, The University of Queensland, St Lucia 4072, Australia; 4Department of Medicine, The University of Melbourne, Austin Health, Heidelberg 3084, Australia

**Keywords:** type 1 diabetes, dietary AGEs, RAGE, autoimmunity

## Abstract

Type 1 diabetes (T1D) is an autoimmune disease in which the β-cells of the pancreas are attacked by the host’s immune system, ultimately resulting in hyperglycemia. It is a complex multifactorial disease postulated to result from a combination of genetic and environmental factors. In parallel with increasing prevalence of T1D in genetically stable populations, highlighting an environmental component, consumption of advanced glycation end products (AGEs) commonly found in in Western diets has increased significantly over the past decades. AGEs can bind to cell surface receptors including the receptor for advanced glycation end products (RAGE). RAGE has proinflammatory roles including in host–pathogen defense, thereby influencing immune cell behavior and can activate and cause proliferation of immune cells such as islet infiltrating CD8^+^ and CD4^+^ T cells and suppress the activity of T regulatory cells, contributing to β-cell injury and hyperglycemia. Insights from studies of individuals at risk of T1D have demonstrated that progression to symptomatic onset and diagnosis can vary, ranging from months to years, providing a window of opportunity for prevention strategies. Interaction between AGEs and RAGE is believed to be a major environmental risk factor for T1D and targeting the AGE-RAGE axis may act as a potential therapeutic strategy for T1D prevention.

## 1. Introduction

Type 1 diabetes (T1D) is an autoimmune disease that comprises 5–10% of all cases of diabetes globally [1]. It is most commonly diagnosed in younger individuals, although the prevalence of adults diagnosed with T1D has increased significantly over the past decades [2]. Although T1D is the most common form of diabetes in children, the incidence continues to increase by about 2–3% per year globally across all age groups, with the most significant increase in prevalence observed in adults [3,4]. Many therapies targeting inflammatory and immune pathways such as anti-CD3 monoclonal antibodies [5] and IL-1 receptor antagonists [6] have been tested over the past decade but have only been shown to slow disease progression in early stages prior to clinical onset [5,7,8]. Currently, there is no cure for T1D. The only approved first-line therapy for T1D is exogenous insulin, the only way to manage disease symptoms and enable most individuals to lead a relatively healthy and long life. However, variations in blood sugar concentrations can be life-threatening and are associated with increased risk for complications. Approximately 40% of individuals with type 1 experience microvascular and macrovascular complications leading to premature mortality and commonly have ongoing mental health impacts due to complex disease management [9].

Due to the lifelong nature of T1D, there is significant interest in preventing, or delaying, the onset and progression of this disease. Disease pathogenesis occurs over many years prior to diagnosis, suggesting that there could be a therapeutic window of opportunity for prevention before specific clinical diagnosis of “overt” disease. A pathway suggested as important for the onset and progression of T1D is the advanced glycation end products (AGEs) and their receptor, the receptor for advanced glycation end products (RAGE) (AGE-RAGE) axis. This review will discuss the pathogenesis of TID, the contribution of genetic susceptibility and environment, how AGEs and RAGE might be involved and how reducing AGE and RAGE signaling through dietary restriction of AGEs may be crucial for T1D prevention.

## 2. Main Text

### 2.1. Pathophysiology of Type 1 Diabetes

T1D is a chronic autoimmune disease where β-cells in the islets of Langerhans of the pancreas are progressively damaged, leading to a critical loss of insulin production resulting in life-threatening high glucose concentrations in the blood (hyperglycemia; Figure 1) [2]. Individuals with T1D often present with common symptoms such as frequent urination (polyuria), fatigue, and weight loss [10]. In addition, individuals with T1D also present with islet-specific antibodies against self-antigens signifying pancreatic autoimmunity. Although antibodies are not postulated to have pathological roles, they indicate disease progression and the presence of 2 or more islet antibodies confers a significantly increased risk of developing T1D [11]. The most common autoantibodies in both children and adults are against insulin (IAA), glutamic acid decarboxylase (GADA), insulinoma antigen-2 (IA-2A), and zinc transporter 8 (ZnT8A) [12]. However, IAA usually presents early in life while GADA often appears much later in childhood [12]. In addition, autoantibodies are not detectable at any stage in some individuals, which complicates diagnosis of diabetes type [13], particularly in adults.

### 2.2. Genetic Susceptibility

T1D is a very heterogeneous disease with over 90 loci encoded on different regions of the genome implicated in autoimmunity [14]. In particular, genes encoding for human leukocyte antigen (HLA) class II haplotypes, including alleles DR3/4 and DQ8 are most commonly associated with T1D [15]. HLA class I and II molecules are involved in control of self-antigens and recognition of pathogens [15]. The different HLA variants influence the presentation of antigens to T cells and signal transduction post-antigen binding, which alters immune tolerance and likely changes the threshold for reactions to self-antigens leading to autoimmunity [16]. Although genetic factors are important, the increasing prevalence of T1D in genetically stable populations suggests that environmental factors are also necessary for precipitation of disease. The individual contributions of and interaction between genetic and environmental factors in T1D are commonly studied in population-based twin cohorts and diabetic animal models. Studies examining monozygotic and dizygotic twins showed that the majority of twins, approximately 70% were discordant for T1D, further supporting that environmental triggers are essential for T1D development [17,18]. The Non-obese diabetic (NOD) mouse model of autoimmune diabetes has contributed significantly to the understanding of T1D, as the progression of diabetes shows similarity to that in humans [19]. Autoantibodies against “self” islet antigens appear in early life and precede progressive defects in insulin secretion and islet invasion and destruction by cells of the immune system [20]. NOD mice also have genetic susceptibility linked to MHC class II alleles as is seen in humans and the presence of antigen-specific immune cells, especially the CD4^+^ and CD8^+^ T cells in lymphoid organs and the pancreatic islets [21]. Due to the barriers in obtaining human pancreata to study all possible pathogenic mechanisms of T1D, NOD mice present a useful starting point for rationalization of potential therapeutic targets.

### 2.3. Autoimmunity and Role of the Immune System

Tolerance is an essential process in the development of the adaptive immune system where regulated unresponsiveness of the immune system to self-antigens is achieved [22]. In healthy individuals, it is common for some autoreactive T cells to escape central tolerance. Normally, when this occurs, antigen-specific cells recognizing “self”-antigens are usually deleted, neutralized or suppressed by processes such as secondary selection in the peripheral tissues, including at sites such as the local lymph nodes and spleen [23]. However, in individuals with T1D, there is a loss of tolerance to self-antigens due to defects in central and peripheral tolerance, which is elegantly reviewed here [21,24]. The reasons for this are not well understood. Some theories suggest that various environmental factors such as viral infections and dietary factors in combination with genetic risk, can result in inflammation of the pancreas and/or pancreatic islets leading to cell death and presentation of self-islet antigens by MHC molecules to the immune system [10,21,25,26,27]. Hence, the postulate is that in T1D, abnormalities in central and peripheral tolerance then allow antigen specific self-reactive T cells to escape these processes and go on to interact with self-antigens in the pancreas and other sites, leading to autoimmunity and autoantibody production [28].

Development of T1D commonly occurs over several years and is divided into stages (Figure 1) involving both the innate and adaptive immune systems. Stages one and two occur before the clinical diagnosis and hence are termed as “prediabetes” and are detected by the appearance of two or more autoantibodies, followed by dysglycaemia (where blood glucose concentrations are above the normal range following tolerance testing but do not meet the criteria for diagnosis) [29]. The third stage, where the diagnosis is made, is characterized by high blood glucose concentrations (hyperglycemia) and symptomatic onset [30]. It is generally believed that β-cell damage occurs prior to symptomatic onset and autoantibody levels should reduce over time. However, this has been challenged in some studies [31] where there was a positive relationship between T cell autoimmunity to islet antigens and disease duration, suggesting that T1D is very heterogenous and progression of the disease can be dependent on many factors such as age at diagnosis, family history immunogenetic profile and disease duration. Factors such as obesity, BMI and energy intake also appear to influence the progression of type 1 diabetes, with higher BMI and obesity associated with earlier onset of T1D [32,33,34] and progression of islet autoimmunity [35]. This has been attributed to factors such as insulin resistance and chronic inflammation acting as accelerating factors in type 1 diabetes pathogenesis [33,36]. Although clinical management of individuals with T1D continue to improve, individuals diagnosed with T1D are still limited to exogenous insulin therapy, with only a few individuals (18% of children and 13% of adults) achieving the recommended glycemic target of <7% for glycated hemoglobin (HbA1c) [37,38]. Once individuals are diagnosed, there are currently no therapies to reverse β-cell loss nor improve β-cell function. Therefore, prevention strategies for T1D in the early stages are a major goal of current research.

**Figure 1 cells-11-03503-f001:**
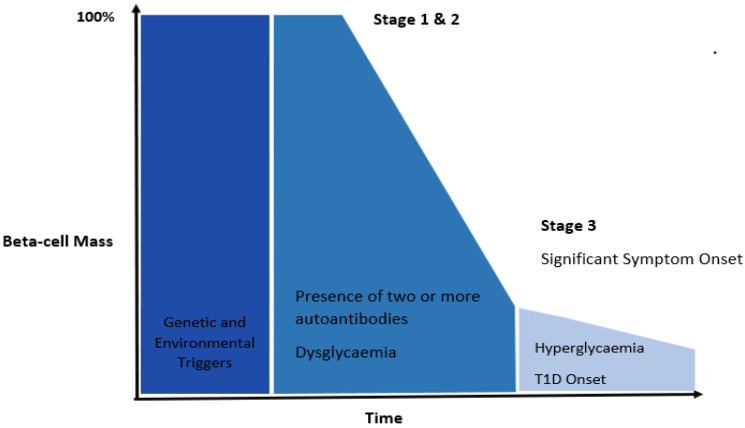
**Stages of T1D according to classical theories.** Genetic susceptibility and environmental factors can influence the onset of T1D. T1D has two preclinical stages which are present before clinical diagnosis. Stage 1 is characterized by the presence of two or more autoantibodies specific to β-cell antigens. People in this stage are generally asymptomatic and have normal glycemic control. Stage 2 involves changes in glycemic control (dysglycemia) measured as a loss in insulin secretion (by C-peptide) during a mixed meal tolerance test in combination with the presence of 2 or more autoantibodies. Stage 3 involves significant clinical symptomatic onset and commonly, people at this stage have severe β-cell loss/damage. Individuals in Stage 3 can present with symptoms such as polyuria, weight loss and fatigue and require exogenous insulin therapies. The progression from Stage 1 to 3 can vary between several months to years since many individual have a so called “honeymoon” period where their pancreas is still able to produce a significant amount of its own insulin reducing exogenous insulin requirements [39].

Early in T1D development in NOD mice, islet inflammation termed insulitis occurs, where infiltration of immune cells such as T and B lymphocytes, macrophages, and dendritic cells (DCs) around and into the Islets of Langerhans is seen [40]. Following the study of human pancreata using resources such as the Juvenile Diabetes Research Foundation (JDRF) Network for Pancreatic Organ Donors with Diabetes (nPOD) [41,42,43], it has been discovered that the number of immune cells infiltrating the islets is often more variable than in mice. Indeed, it is more commonly present in individuals with a younger age of onset and with short disease duration [44]. As suggested above, the loss of β-cell specific tolerance can lead to β-cell damage and destruction by immune cells, facilitating the release of self-antigens by the pancreas. These are endocytosed by antigen-presenting cells (APCs) such as resident DCs and macrophages. Several studies have supported the role of both these cell types in the precipitation of T1D. When macrophages phagocytose antigens or cell fragments, activation of the innate immune system likely triggers inflammatory pathways, further exacerbating damage to β-cells and potentially recruiting the adaptive immune system [45,46,47,48]. Whether this process is an early event, occurring prediabetes in response to an environmental trigger such as a viral infection or involved in perpetuation of immune cell damage to β-cells in the longer term remains to be determined. In addition to their role in antigen presentation, dendritic cells (DCs) also play important roles in maintaining tolerance through the suppression of auto-reactive T cells and induction of Tregs [49,50]. This is not discussed further in this review but has been elegantly summarized previously [51,52]. In T1D pathogenesis, the perpetuation of self-antigen presentation and inflammation due to aberrant macrophage and DC function have been identified both in murine models [51,53,54] and in individuals who later develop diabetes [55,56,57]. Resident DCs in the pancreas also recognize danger signals from the damaged β-cells, leading to the release of chemokines which attract other immune cells. DCs then migrate to the pancreatic lymph node, presenting β-cell antigens to T cells, thereby activating effector CD4^+^ and CD8^+^ T cells (Figure 2) [58]. It is also postulated that DCs can do this in situ in the pancreas [59].

Many studies have shown that T cells (Figure 2) play an essential role in both the progression and development of T1D [41,48,60]. CD4^+^ T cells can secrete proinflammatory cytokines, attracting additional T and B cells and other immune cells to surround or enter the pancreatic islets [61]. CD4^+^ T cells also provide help to support effector CD8^+^ T cell activation which are postulated as the β-cell assassins in T1D. CD8^+^ T cells, generally mediate β-cell death directly via perforins, granzymes and FAS-FAS ligand death pathways [61,62]. Studies in NOD mice have shown that T1D can only be developed in the presence of both CD4^+^ and CD8^+^ T cells, but not by either population alone [60]. An early study found that anti-CD3 monoclonal antibodies act as immunosuppressants by reducing T cell activation and can prevent T1D onset in NOD mice [63]. This has led to the development of a promising therapy using the anti-CD3 monoclonal antibody, teplizumab in humans, the first drug that has been shown to delay T1D diagnosis in individuals at high risk (defined as having more than two autoantibodies and relatives of individuals with T1D) [5]. In humans, studies of pancreatic sections show significant infiltration by CD4^+^ and CD8^+^ T cells [41,64]. In an early study, increases in both CD8^+^ T cells and MHC class I expression were seen in individuals newly diagnosed with T1D [65]. Studies have also observed the recurrence of T1D in recipients who received pancreas transplantation from an HLA identical donor, with immune infiltration consisting predominantly of CD8^+^ T cells, a few CD4^+^ T cells and near-total destruction of β-cells [66]. Taken together, these studies further highlight the importance of T cells in T1D development and onset, and provide evidence that CD8^+^ T cells may be the primary effector immune cells responsible for the destruction of β-cells in T1D.

Studies of immune infiltration in T1D have mostly focused on infiltration into the islets or the endocrine compartment of the pancreas, and often the areas surrounding the islets, but the exocrine structures are often overlooked in T1D development. However, it has become increasingly clear that individuals with T1D also have exocrine insufficiency. The major function of the exocrine pancreas is the production of digestive enzymes. Studies have shown that in individuals with T1D, the exocrine pancreas is reduced by approximately 30% [67]. Studies in the human pancreas have also observed more DCs and T cell infiltration into the exocrine pancreas in organ donors with T1D [68]. These findings suggest that there may be pathological connections between the endocrine and exocrine pancreas. Further understanding of immune cell behavior and accumulation in the exocrine pancreas may provide new insights into the reasons for islet damage and infiltration, and assist in the generation of effective and long-lasting therapies to prevent T1D onset.

Another key component of peripheral immune regulation involves regulatory T cells (Tregs). Forkhead box P3 protein (FOXP3) positive Tregs are regulatory T cells responsible for maintaining homeostasis by suppressing the activity of effector T cells, including CD4^+^ and CD8^+^ T cells. Increased inflammatory cytokine production by effector T cells may alter the phenotype of Tregs, reducing their suppressive abilities [69]. In humans, a rare disorder known as the immunodysregulation polyendocrinopathy enteropathy X-linked syndrome (IPEX), due to the loss of function of the *FOXP3* gene, can lead to the onset of many autoimmune diseases, especially T1D, in which is diagnosed in >80% of affected individuals before two years of age [70]. This highlights the importance of Tregs in maintaining tolerance, and the importance of functional Tregs in preventing T1D development. Imbalances between regulatory and effector T cells is thought to contribute to autoimmunity and T1D development [69], and individuals are often reported with a deficiency in interleukin 2 (IL-2) production [71]. IL-2 is arguably the most important growth factor for T regulatory cell function and increases the expression of FOXP3 and Treg suppression molecules with the ability to inhibit the activation of perforin or granzymes released from killer CD8^+^ T cells, limiting autoimmunity [72]. Studies found that a low-dose IL-2 injection can stimulate pancreatic Treg function and proliferation and reduce production of the proinflammatory cytokines such as interferon-gamma (IFN-γ) [71]. Increasing Treg cells in the pancreas by administration of IL-2 has been found to reverse disease in NOD mice. Several clinical studies have found that although frequent low-dose IL-2 therapy increased Tregs population, they were transient and returned to pretreatment levels after treatment [73]. Another study examined a slightly higher IL-2 dose and observed an increased expansion of non-Tregs such as Natural Killer (NK) cell populations, potentially shifting from immune tolerance to activation [74]. The off-target effects limit the therapeutic potential of IL-2 in individuals at high risk and established T1D [71]. Although T cells are pivotal for T1D development, it is postulated that other immune cells such as B cells and NK cells [75] also have important roles in the pathogenesis of T1D. These cells’ involvement in T1D is not discussed further here but elegantly reviewed in Lehuen et al. (2010) [48].

### 2.4. Environmental Triggers, Dietary AGEs, Inflammation and T1D Risk

Several environmental factors have been postulated to trigger inflammation and potentially β-cell autoimmunity in T1D. Enteroviral infection is one of the most well studied environmental factors and is associated with insulitis and β-cell loss, in particular those of the Coxsackievirus B family. Enteroviruses elicit strong immune responses by targeting β-cells surface receptors, inducing inflammation and promoting autoimmunity [60]. Several human prospective studies [76,77] have observed positive associations between enteroviral infections and progression to autoimmunity and clinical diagnosis of T1D, however, there are also studies that have found no association [78]. With the recent COVID-19 pandemic, the incidence of T1D appeared to be higher than in previous years with increases in both incidence and severity at diagnosis, measured as diabetic ketoacidosis [79,80]. Although clearly implicated in disease pathogenesis, more research is required to elucidate the mechanisms by which viral infections induce β-cells specific autoimmunity and damage. Other environmental factors suspected to contribute to alter risk for T1D development include vaccinations, lower vitamin D levels, and exposure to toxins. These are not further discussed here but are elegantly reviewed in Rewers & Ludvigsson (2016) [25].

Another major environmental factor which may influence the risk for T1D is increased consumption of highly processed foods. Food processing imparts commercially desirable properties such as increased shelf-life, flavor and digestibility, using methods such as dry heat baking [81]. This, in combination with the higher sugar, fat, refined carbohydrate and lower wholegrain content of highly processed foods, amplifies the production of chemical compounds known as advanced glycation end products (AGEs). The most common and well-studied AGE ligands in vivo are carboxymethyl lysine (CML) and methyl-glyoxal derived hydroimidazolones (MG-H1), which are both known to bind to AGE receptors such as RAGE [82]. Scheijen et al. (2016) found that food items with short heating times that are low in protein and carbohydrates and high in water, such as fruits and vegetables, have the lowest AGE content [83]. AGEs are formed via non-enzymatic reactions, where reducing sugars such as glucose react with amine residues on proteins [84] (reviewed in detail here, [85]). This process involves the Maillard reaction which can be subdivided into three steps (Figure 3), with the first step producing Schiff bases before rearranging into more stable Amadori products which are reversible, followed by the formation of AGEs which are irreversible [86]. Exogenous AGEs in foods can be absorbed across the gastrointestinal tract, contributing to the body’s AGE pool [87]. Based on human and animal studies, approximately 10% of total AGEs eaten are thought to be absorbed into the blood, with two-thirds remaining in the body for more than three days [88]. Studies in mice have confirmed that consumption of dietary AGEs can elevate serum and tissue AGE levels [81].

In addition to ingestion from the diet, AGEs are also formed endogenously in the body, an example of which is HbA1c and fructosamine albumin, used clinically to track long term blood sugar control. Endogenous AGEs are produced as a natural consequence of cellular metabolism, particularly in environments with high oxidative stress [89]. In diabetes, the abundance of glucose in the bloodstream accelerates and exacerbates the formation of AGEs [90]. Clearance of AGEs from the body occurs via the gastrointestinal tract (GIT) and liver into the feces, and via the kidneys into the urine. As such, the AGE content in the body at any time is influenced by the amount of AGEs formed endogenously, AGEs ingested, their biodistribution, metabolism and the rate of excretion into the urine and feces [91]. Dietary AGE digestion and absorption in the GIT may interact with and change the composition of gut microbiome, as the gut microbiome is largely influenced by diet. The gut microbiome has increasingly been recognized as one of the key factors contributing to T1D onset by altering gut permeability and immune homeostasis [92]. Many studies in vitro and in vivo have observed that high dietary AGEs can alter the microbial composition, though the specific microbial changes require further validation [93,94,95]. Microbial dysbiosis can mediate immunological responses resulting in β-cell damage and T1D onset. The impact of AGEs on gut homeostasis is not further discussed, but is reviewed in Zhou et al. (2020) [96] and Snelson & Coughlan (2019) [97].

AGE research over the past decades has focused mainly on their role in the development and progression of various chronic complications of diabetes [98]. Due to the long half-life and abundance of plasma proteins and components of the extracellular matrix (ECM) including albumin, fibrinogen, and collagen, they are the predominant targets for glycation [84]. Increased glycation and accumulation of AGE modified proteins has been implicated in both microvascular and macrovascular complications of diabetes including kidney disease, retinopathy, cardiovascular disease and wound healing [98]. AGE accumulation on long lived proteins such as collagen in the skin have also emerged as biomarkers for the development of diabetic complications. AGE skin autofluorescence is a non-invasive, fast, and reliable method of measuring AGEs which has excellent associations to actual measured skin and body AGE accumulation, and is reviewed elegantly here [99]. In general, studies show that accumulation and/or higher AGE concentrations are associated with the development or presence of one or more diabetes complications.

Recent studies have highlighted the importance of dietary consumption of AGEs as risk factors for T1D onset. Firstly, high circulating concentrations of the AGE are an important environmental risk factor for T1D when added to the known risk factor, the presence of autoantibodies, to help identify individuals who are most likely to develop the disease [100]. Murine studies have suggested that AGEs initiate β-cell dysfunction via ligation to their receptor RAGE resulting in oxidative stress and insulin deficiency [101]. Studies in isolated islets have found that chronic administration of AGEs reduced insulin content within the islet and insulin secretion in response to glucose stimulation and increased infiltration of immune cells into the pancreatic islets which are via effects on mitochondrial superoxide production and ATP production [101,102]. High AGEs were also found to inhibit ATP synthesis, inhibiting the ATP-dependent K+ channel from opening, leading to impaired glucose-stimulated insulin secretion [102]. Here, AGEs impaired insulin secretion by elevating nitric oxide synthase (NOS) and thus nitric oxide concentrations in the pancreatic islets and INS cells. This subsequently resulted in inhibition of electron transport chain component cytochrome c oxidase, resulting in reduced ATP synthesis and reduced insulin secretion [102]. When healthy rats were exposed chronically to a high AGE diet, insulin secretory defects and islet infiltration in the absence of diabetes were observed [101]. Similarly, increased islet hormone content including insulin, proinsulin, and glucagon, and reduced immune infiltration were observed in mice fed with a low AGE diet compared to a high AGE diet [103,104]. A study in humans found that a higher serum AGE concentration was seen in at risk children who progressed to T1D compared to non-progressors [101]. This suggests that increased AGE consumption can impair β-cell function contributing to T1D development.

In addition, other studies have shown that the effects of dietary AGEs are not limited to postnatally, but also affect β-cell function during the perinatal and gestational period. Maternal-blood and food-derived AGE burden in early life are reported to prematurely raise babies’ circulating AGE concentrations, influencing their risk for inflammatory conditions such as T1D [105]. Studies using dietary AGE restriction in NOD mouse models have found that lowering consumption of dietary AGEs in mothers from conception to weaning significantly decreased fasting blood glucose concentrations and improved insulin responses to glucose in offspring [103,104]. Further, pancreatic islets from offspring mice where the mothers consumed a high AGE diet during and post gestation showed increased immune cell infiltration compared to those infants whose mothers consumed a low AGE diet [103]. Importantly, intergenerational dietary AGE study observed a significant reduction of T1D incidence in mice fed with the low AGE diet compared to the high AGEs diet [104]. Further, inflammatory changes induced by high dietary AGE consumption can be ameliorated by AGEs lowering therapy alagebrium chloride [101,106].

Finally, in healthy individuals, there is a positive relationship between the circulating concentrations of AGEs and insulin secretion [107]. However, the rate of AGEs clearance is often greatly altered in individuals with diabetes who also have impairments in either kidney or liver function influencing insulin secretion and sensitivity [108]. Increased prevalence of obesity in individuals with T1D could also increase the likelihood of developing liver diseases such as Non-alcoholic steatohepatitis (NASH) and Non-alcoholic fatty liver disease (NAFLD) [109,110], impacting AGEs clearance. Obesity has been increasingly recognized to be associated with earlier onset of T1D and development of a “T1D like” phenotype characterized by immune infiltration and islet autoimmunity [36]. Perhaps not surprising is that lower dietary AGE intake improves insulin sensitivity in overweight individuals [111,112] as well as decreasing chronic low-grade inflammation. Low AGE diets also impact renal function in overweight and obese individuals [113] and hepatic function in rodent models of obesity and liver fibrosis [114,115]. Obesity has long been recognized as a risk factor for type 2 diabetes (T2D) but the increasing prevalence of obesity in individuals at risk for and diagnosed with T1D draws highlights the need for therapies that not only lower blood glucose but also provide greater weight management. Taken together, these findings suggest that increasing AGEs concentrations in the body can lead to β-cell damage and impaired β-cell function, possibly by influencing immunoregulation, contributing to T1D. Although the loss of β-cells through autoimmunity is unique in T1D, the mechanisms by which dietary AGEs induce β-cells dysfunction could also apply to T2D, but this is not further discussed in this review.

### 2.5. AGE Binding to RAGE

RAGE is a member of the immunoglobulin superfamily encoded by the *AGER* gene [116]. It is a proinflammatory receptor expressed in many cell types including immune cells and is important for host–pathogen defense [117]. AGEs bind to the cell surface AGE receptor RAGE [118] and upregulate its expression, a process exacerbated in diabetes via a positive-feedback loop [119]. However, RAGE is a multiligand receptor which can be stimulated by various ligands other than AGEs, including amyloid-B, amphoterin, and S100-calgranulins (Figure 3) [84]. Compared to other ligands of RAGE, AGEs have a relatively low affinity, and at low serum concentrations are unlikely to induce strong RAGE signaling [120]. However, in diabetes, AGEs concentration increases significantly due to the abundance of glucose in the bloodstream, therefore significantly upregulating RAGE signaling [116].

Another group of receptors known as the scavenger receptors also exhibit binding affinity for AGEs. Scavenger receptors are also multiligand receptors expressed on different cell types including the endothelial cells and immune cells such as macrophages. Research into scavenger receptor proteins has mainly been focused on the link to macrovascular and microvascular complications of diabetes, in particular atherosclerosis [121]. For example, class B scavenger receptor SR-B1 plays an important atheroprotective role, binding cholesteryl esters from high density lipoprotein (HDL-CE) and eliminating excessive cholesterol from peripheral tissues through cholesterol efflux [122]. However, in vitro AGEs inhibit SR-B1 selective cholesterol efflux to HDL [123] and HDL-CE uptake, suggesting pathological roles of AGEs binding to scavenger receptors. Similarly, the SR-B family receptor CD36 also binds AGEs [121,124] and has garnered substantial interest in the context of diabetes and its complications [125]. Like RAGE, CD36 is expressed on a wide variety of cells, including pancreatic β-cells, monocytes, macrophages and subsets of T and B cells [126] and binds a wide variety of ligands. CD36 and associated downstream signaling has been implicated in the development of insulin resistance, β-cell dysfunction and loss of β-cell mass in metabolic syndrome and type 2 diabetes (reviewed here [125]). Further CD36 has been implicated in a reduction in Treg and CD8+ T cell functionality in the tumor microenvironment [127,128], supporting tumor growth, although these findings appear to relate to CD36′s role as a lipid transporter [126]. Whether CD36 is involved in T1D pathogenesis remains to be seen. Galectin 3, another AGE binding scavenger receptor [129], is widely expressed immune cells where it can activate proinflammatory pathways including cytokine and chemokine release [130]. Encoded by LGALS3, there is some evidence to suggest LGALS3 is a T1D susceptibility gene [131] and this has been supported by a small number of in vitro and in vivo studies suggesting a role for galectin 3 in cytokine mediated β-cells apoptosis and survival [131,132,133]. More research into AGEs and scavenger receptor family is required to understand whether these receptors are involved in the modulation of immune activity in T1D and whether they have similar roles to RAGE in T1D development.

Despite AGE and RAGE interactions being studied for more than 20 years, most studies have focused on the influence of AGE-RAGE binding on pathological conditions. However, AGEs are produced under physiological conditions where their role is poorly understood and it is not clear whether varying AGE modifications can elicit different physiological responses via receptor signaling, such as via RAGE. Certainly, binding of AGEs to RAGE initiates intracellular signaling pathways leading to inflammation and oxidative stress which are important part of host–pathogen defense, but can also result in β-cell injury [101,106,108]. However, AGE ligation resulting in RAGE signaling is also implicated in many other inflammatory conditions including neurodegenerative conditions, rheumatoid arthritis, and inflammatory bowel disease [82,98,134,135]. The binding of AGEs to RAGE commonly activates the nuclear factor kappa B (NF-κB) inflammatory pathway and other transcription factors such as mitogen activated protein kinase (MAPK), and Janus kinase (JAK-STAT), increasing the production of proinflammatory cytokines (Figure 4) [136].

Circulating RAGE isoforms lacking the transmembrane domain, collectively termed soluble RAGE (sRAGE), are also present in all mammals (Figure 4). These are comprised of the endogenously secreted esRAGE and a cleaved isoform derived from membrane bound RAGE [137]. The function of sRAGE is not fully understood but is thought to have anti-inflammatory properties by competing with membrane bound RAGE for ligands [138,139]. Salonen et al. (2016) [140] found that sRAGE concentrations are reduced during seroconversion to positivity for islet autoantibodies in children who later progressed to T1D. Studies in children at high risk for developing T1D have also identified functional polymorphisms of the RAGE gene (*AGER*) which impact circulating sRAGE concentrations and significantly alter the risk for T1D development [106,141]. Lower plasma sRAGE concentrations in early life are also associated with a more severe presentation of T1D at diagnosis including diabetic ketoacidosis [142]. At the point of diagnosis and beyond, higher sRAGE concentrations are associated with the presence of microvascular and macrovascular complications and greater mortality in adults with T1D [143]. Taken together, these studies suggest that high sRAGE may be triggered by activation of the AGE-RAGE axis and act to counter chronic inflammation.

RAGE is also expressed by many of the immune cells involved in the pathogenesis of T1D such as dendritic cells, macrophages and T and B lymphocytes, and this appears to be linked to their function and activity. A reduction of RAGE expression on T cells significantly reduces proinflammatory cytokines such as IFN-γ responsible for islet inflammation [144] and influences T regulatory cell function [145]. Chen et al. (2008) [146] found that antagonism of RAGE with a small molecule inhibitor TTP488 reduced the destruction of islets transplanted into NOD mice with diabetes, delaying islet rejection. A study by Han et al. (2014) demonstrated that AGEs promote RAGE expression on CD4^+^ T cells and differentiation to proinflammatory CD4^+^ T cell subsets. This was prevented by the knockdown of RAGE [147]. These data suggest that RAGE signaling may alter T cell phenotype impacting autoimmunity and inflammation, as seen in T1D. Studies using human T cells cultured from peripheral blood mononuclear cells (PBMC) found that RAGE expression on CD4^+^ and CD8^+^ T cells in participants with normal blood glucose was higher in participants who progressed into T1D than those who did not. Culturing human CD8^+^ T cells with and without RAGE ligands found that RAGE expression was significantly increased in T cells exposed to RAGE ligands [148]. These findings suggest that increased RAGE expression on T cells occurs prior to T1D onset [149] and that increased consumption of AGEs can enhance RAGE expression on T cells, modulating T cells behavior [146,148].

Although increased dietary AGEs consumption has been shown to be associated with β-cell injury and defective insulin secretion in mice via the AGE and RAGE pathway, the precise mechanisms in which dietary AGEs interact with RAGE remain to be elucidated. In mice, RAGE receptors are found to be expressed on the cell surface of APCs and CD4^+^ and CD8^+^ T cells. However, a study examining RAGE expression in human T cells found that RAGE is expressed exclusively intracellularly [148]. Therefore, more research is required to understand the interaction between dietary AGEs which are found extracellularly, and RAGE receptors which are expressed both on the cell surface and intracellularly depending on the cell type. Therapeutically, only a few studies examined pathways to block RAGE signaling in T1D. A recent study in NOD mice has clearly shown that increasing sRAGE through intraperitoneal administration of human recombinant sRAGE can reduce AGE burden, improve insulin expression and delay T1D onset via improvements in Treg function [145]. Another study by Zhang et al. (2020) examined the use of inhibitors to decrease the accumulation of other RAGE ligands such as High Mobility Group Box Protein -1 (HMGB-1), which was found to delay the onset of T1D in mouse models [150]. Indeed, activation of RAGE by binding HMGB-1 acts as an immune mediator to enhance autoimmune progression and diabetes onset in NOD mice [151]. These findings provide further evidence that interruptions in the dietary AGE-RAGE axis might be key in managing immunoinflammatory diseases such as T1D.

## 3. Conclusions

Although the role of AGEs and RAGE in T1D remains to be fully understood, there is clear evidence suggesting that this axis is involved in both the development and progression of T1D. AGEs interact with the immunological receptor RAGE to impact immune function and inflammation. Further, consumption of AGEs alters β-cells function and risk factors for T1D such as insulin sensitivity and obesity. Lowering of dietary AGE intake or interruption of AGE-RAGE binding reduces the incidence of T1D in murine models. Meanwhile, population-based studies have also identified that increases in circulating AGEs increases risk for T1D development in childhood and adolescence. Given that AGEs are regularly consumed as part of Western diets, as well as being made endogenously when sugar concentrations are elevated in the body, AGEs are important modifiable environmental risk factors for T1D. Hence, modifying dietary AGEs represents a simple and risk-free approach that should be further investigated in animal and clinical studies to examine the impacts on β-cell function, autoimmunity and T1D development.

## Figures and Tables

**Figure 2 cells-11-03503-f002:**
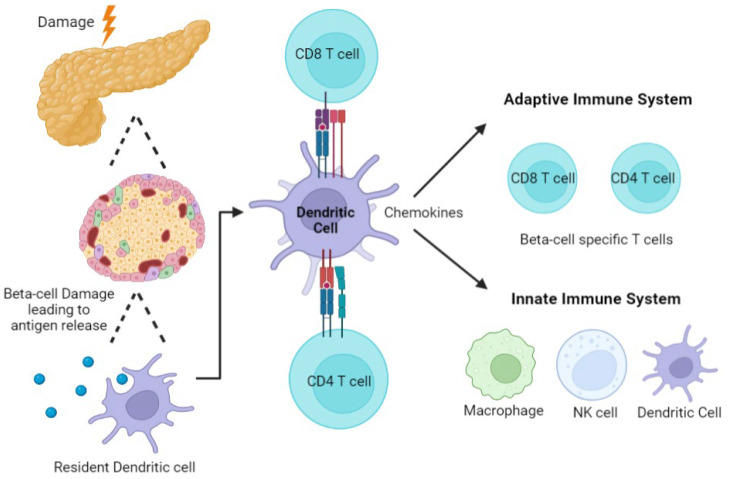
**Major Immune Cells Implicated in T1D Pathogenesis.** Damage to the β-cells of the pancreas can lead to the release of antigens. Antigens can be recognized by resident DCs which leads to the uptake and presentation of islet specific antigens to T cell. DCs will present antigens to T cells in the pancreatic lymph node leading to the release of chemokines which attract additional β-cell specific T cells and innate immune cells. If these β-cell autoantigen specific T cells are not removed by peripheral tolerance, they can further enhance inflammation and contribute to β-cell death.

**Figure 3 cells-11-03503-f003:**
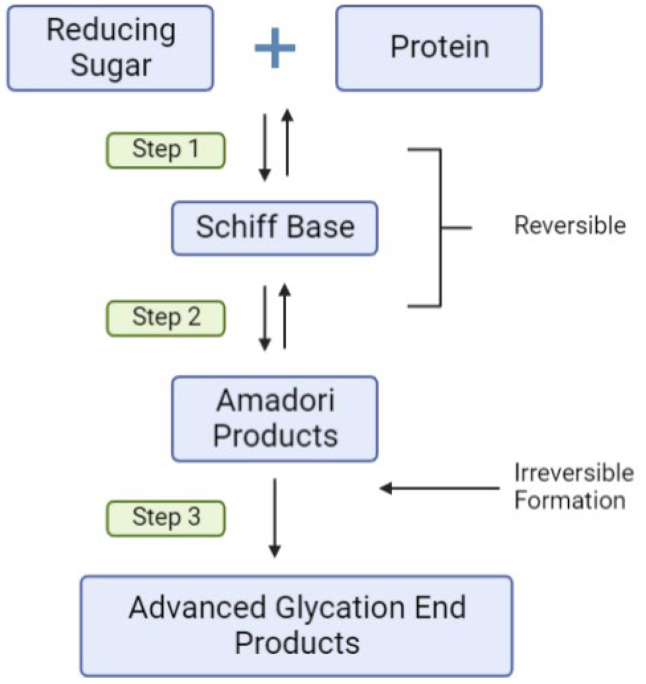
**The basic chemistry of AGE Formation.** Reducing sugars such as D-glucose, fructose, ribose and reactive dicarbonyls, including methylglyoxal, interact with amine groups such as lysine, tyrosine and arginine on proteins and peptides. Following various reversible chemical transition stages, the irreversible formation of advanced glycation end products such as Nε carboxymethyllysine modifications, HbA_1C_, fructosamine albumin and methylglyoxal hydroimidazolones occurs.

**Figure 4 cells-11-03503-f004:**
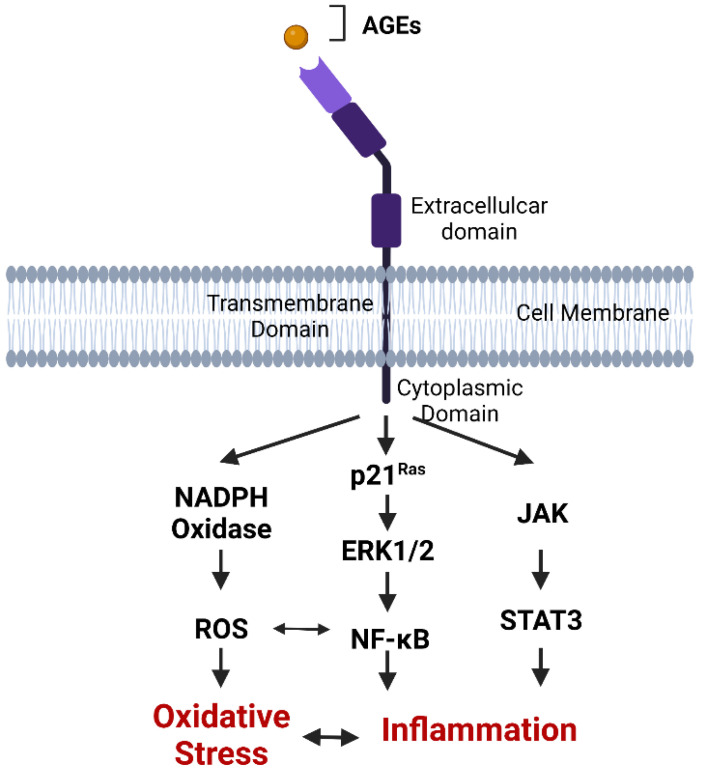
**AGE-RAGE signaling.** RAGE is composed of three extracellular domains, which include a V type ligand-binding domain and two C domains, one transmembrane domain and a cytoplasmic domain essential for signal transduction. RAGE ligands such as AGE bind to the V domain of RAGE on the cell membrane, leading to the initiation of inflammatory pathways such as JAK-STAT, NADPH Oxidase- NF-κB, p21^Ras^ and production of reactive oxygen species resulting in β-cell damage.

## Data Availability

Not applicable.

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
