# Peer review of "Advanced Glycation End Products and Inflammation in Type 1 Diabetes Development"

_cells, 2022, doi:10.3390/cells11213503_

Round 1
Reviewer 1 Report
Du et al aimed to review the role of advanced glycation end products (AGE) and inflammation in T1D. The topic of the review is timely and of interest. The authors review a very large number of relevant studies. However, in its present form, it contains inaccuracies and lack of clarity in some critical areas.
Specific examples of the above are the following, ordered by appearance, not by importance:
- Abstract:
o I recommend adding “ultimately” before the word “resulting in hyperglycemia” since many important steps are obviated in this sentence.
o “suggesting an environmental factor” added after “stable populations,” would help clarify the meaning of the sentence.
o Clarifying the mechanisms that the authors allude to in “diabetogenic mechanisms” would be important since it is the key of the review.
o Consider adding “for T1D” after “risk factor” in the last sentence.
- Introduction:
o T1D is widely accepted to be an autoimmune disease but the inflammatory etiology is less established. Since this is one of the keys of the review, it requires clear explanations and solid references to support this concept. I recommend removing this word from the very first sentence in the article and, instead, explaining this concept later, when the concept of inflammation is introduced.
o T1D is the most common form of diabetes in children. However, it continues to be diagnosed throughout life and, ultimately, it appears that more cases are diagnosed in adult life than during childhood. Therefore, the second sentence is inaccurate.
o “without evidence of sustained benefits to prevent disease onset in humans” is inaccurate since Teplizumab has shown benefit, as mentioned by the authors later in the review.
o The two references supporting the above statement are very outdated, given the many recent advances in this area. Also, if the authors mention anti-inflammatory treatments, an appropriate reference should be provided.
o Microvascular complications also happen, in addition to macro- and mental.
- Figures are not cited in the text
- Pathophysiology of type 1 diabetes:
o Autoantibodies to GAD, insulin, IA-2 and ZnT8 may be present not only in children but in adults as well.
o The last sentence is overall inaccurate. It may be more accurate to say that autoantibodies are not detectable in some individuals, which complicates diagnosis of diabetes type. HLA is not relevant in this sentence because it is not usually tested in the clinical setting.
- Genetic susceptibility:
o It’s now over 90 loci involved in T1D
o Most, but not all, are involved in the immune response.
o DR3 is in linkage disequilibrium with DQ2 and DR4, with DQ8.
o The major goal of twin studies is evaluate the contribution of genetic and environment, not always and not only interaction.
o Reference 14 is inaccurate. The data in the review corresponds to Redondo et al, NEJM.
o MHC in humans is called HLA
- Autoimmunity
o Please clarify if escaping central tolerance is common in healthy individuals or in T1D
o Please clarify why the presentation of islet antigens released after beta cell death is inappropriate.
o Please expand and clarify the environmental factors that result in pancreatic inflammation since this is relevant to the topic of the review.
o The description of stages 1, 2 and 3 of T1D is wrong in the text and in the figure. Please review those definitions. Some examples are: stage 1 is defined by the presence of two or more positive islet autoantibodies (not only insulin autoantibodies); stage 2 by the addition of dysglycemia (glucose that exceeds normal levels but does not meet the criteria for diabetes) (beta-cell function is not used to define it; symptoms do not usually occur here as opposed to what is indicated in the figure); stage 3 by crossing glycemic thresholds for diabetes and it is usually accompanied by symptoms.
o What do the authors mean by “sustained positive relationship between diabetes duration and […] beta-cell function after diagnosis”? It is the opposite.
o Most, not “some” individuals experience honeymoon or partial remission period.
o It is both younger age at onset and shorter diabetes duration that associated with increased insulitis.
o What do the authors mean by “which may reflect the greater time elapsed…” in nPOD donors?
- Similarly to the overview that is given on genetics and etiopathogenesis of T1D, the authors should include an overeview on environmental factors that have been associated with T1D. Enteroviruses, in particular, have been quite robustly associated.
- Obesity has been associated with earlier onset of T1D and modification of the progression of islet autoimmunity. Given the relationship between AGEs and obesity, adding this piece would strengthen the review.
- Dietary AGEs:
o Please clarify how dry heating baking contributes to AGE formation, and how short heating times lower them.
o I recommend a figure to explain AGE formation since this is critical to this review.
o Kidney disease is not the only complication of diabetes that has been associated with AGEs.
o Please clarify how high AGEs interfere with ATP synthesis
o “Islet hormone” should be plural
o Overall, there is a lack of clarity on the mechanisms that underlie the involvement of AGE in causing diabetes. It would be helpful to list the mechanisms that have been involved in causing diabetes. Beta-cell dysfunction and beta-cell loss due to autoimmune destruction are usually thought of as two separate mechanisms. Please explain the involvement of AGEs on those two separately.
o While autoimmune loss of beta-cells is unique to type 1 diabetes, beta-cell dysfunction could happen in T2D as well. Please clarify if the mechanisms above also influence T2D.
o Not all individuals with diabetes have kidney or liver impairment. It would be better “individuals with diabetes who have kidney impairment”. Also, liver impairment is not characteristic of T1D. It appears in T2D in large part due to the association with obesity and NASH and NAFLD. Please clarify.
o “when added to the presence of autoantibodies”: But autoantibodies are not causal in diabetes. Also, does this imply that AGE could be causing autoantibodies? Please rephrase.
- AGE bidning RAGE
o The first sentence in the second paragraph seems to belong in the first paragraph. Its meaning is unclear in the current position.
o “more severe presentation of T1D at diagnosis including DKA and autoantibody status”: autoantibody status is not a sign of severity in T1D. Please clarify.
o “high sRAGE may be associated with elevated activation of the AGE-RAGE axis”: Could “triggered” be more informative than “associated” ?
o It is unclear why the paragraph “Therapeutically,…” is separated from the previous paragraph that describes interventions to disrupt this axis. Please consider merging those two paragraphs or clarifying why they are separate.
- Figure 1:
o Legend is all wrong (please see previous text): The description of the stages of preclinical T1D as presented here is highly inaccurate.
o Stages 1 and 2 by definition do not have symptoms.
- Figure 2:
o Legend: Review English language please. Also. Does “Made in Bierender” need to be in the legend?
- Conclusions
The present conclusion is too vague to be of use. If there a conclusion, please use it to provide a summary, including an overview of the system, list of mechanisms, steps potentially amenable to intervention, etc
Author Response
Thank you for your careful consideration of and helpful suggestions for the improvement of this manuscript. Please find out itemised responses below.
Abstract:
I recommend adding “ultimately” before the word “resulting in hyperglycemia” since many important steps are obviated in this sentence.
We agree and this has been amended accordingly.
“suggesting an environmental factor” added after “stable populations,” would help clarify the meaning of the sentence.
We completely agree and this has been amended.
Clarifying the mechanisms that the authors allude to in “diabetogenic mechanisms” would be important since it is the key of the review.
Thankyou for this suggestions we have modified the manuscript to clarify the diabetogenic mechanisms that we were referring to.
Consider adding “for T1D” after “risk factor” in the last sentence.
Done.
T1D is widely accepted to be an autoimmune disease but the inflammatory etiology is less established. Since this is one of the keys of the review, it requires clear explanations and solid references to support this concept. I recommend removing this word from the very first sentence in the article and, instead, explaining this concept later, when the concept of inflammation is introduced.
Done.
Introduction:
T1D is the most common form of diabetes in children. However, it continues to be diagnosed throughout life and, ultimately it appears that more cases are diagnosed in adult life than during childhood. Therefore, the second sentence is inaccurate.
While we completely agree that there are more adults globally with diagnosed Type 1 diabetes and hence greater prevalence, the incidence is higher in childhood and adolescence. Please see pages 43/44 and 48 of the current International Diabetes Federation Atlas 2021. We therefore propose the following sentence which we have inserted into the manuscript.
“Although T1D is the most common form of diabetes in children, the incidence continues to increase by about 2-3% per year globally across all age groups, with the most significant increase in prevalence observed in adulthood [3,4].”
“without evidence of sustained benefits to prevent disease onset in humans” is inaccurate since Teplizumab has shown benefit, as mentioned by the authors later in the review.
We agree and have amended this sentence.
The two references supporting the above statement are very outdated, given the many recent advances in this area. Also, if the authors mention anti-inflammatory treatments, an appropriate reference should be provided.
We have added more recent references.
Microvascular complications also happen, in addition to macro- and mental.
Agreed. This has been amended.
Figures are not cited in the text
We apologise and this has been amended.
Pathophysiology of type 1 diabetes:
Autoantibodies to GAD, insulin, IA-2 and ZnT8 may be present not only in children but in adults as well.
The last sentence is overall inaccurate. It may be more accurate to say that autoantibodies are not detectable in some individuals, which complicates diagnosis of diabetes type. HLA is not relevant in this sentence because it is not usually tested in the clinical setting.
Thank you. We agree. We have amended this sentence.
Genetic susceptibility:
It’s now over 90 loci involved in T1D. Most, but not all, are involved in the immune response. DR3 is in linkage disequilibrium with DQ2 and DR4, with DQ8.The major goal of twin studies is evaluate the contribution of genetic and environment, not always and not only interaction.
Thank you. We have included these comments and/or amended the previous text.
Reference 14 is inaccurate. The data in the review corresponds to Redondo et al, NEJM.
The manuscript has been amended accordingly.
MHC in humans is called HLA
Duly noted and corrected.
Autoimmunity
Please clarify if escaping central tolerance is common in healthy individuals or in T1D.
Done.
Please clarify why the presentation of islet antigens released after beta cell death is inappropriate.
Please expand and clarify the environmental factors that result in pancreatic inflammation since this is relevant to the topic of the review.
The description of stages 1, 2 and 3 of T1D is wrong in the text and in the figure. Please review those definitions. Some examples are: stage 1 is defined by the presence of two or more positive islet autoantibodies (not only insulin autoantibodies); stage 2 by the addition of dysglycemia (glucose that exceeds normal levels but does not meet the criteria for diabetes) (beta-cell function is not used to define it; symptoms do not usually occur here as opposed to what is indicated in the figure); stage 3 by crossing glycemic thresholds for diabetes and it is usually accompanied by symptoms.
We agree and these have been clarified.
What do the authors mean by “sustained positive relationship between diabetes duration and […] beta-cell function after diagnosis”? It is the opposite.
These statements have been clarified.
Most, not “some” individuals experience honeymoon or partial remission period.
We have amended this word.
It is both younger age at onset and shorter diabetes duration that associated with increased insulitis.
Thank you. Fixed.
What do the authors mean by “which may reflect the greater time elapsed…” in nPOD donors?
This was meant to reflect that many of the donors in nPOD were older and had a longer diabetes duration which is now addressed by the comment above.
Similarly to the overview that is given on genetics and etiopathogenesis of T1D, the authors should include an overeview on environmental factors that have been associated with T1D. Enteroviruses, in particular, have been quite robustly associated.
We completely agree and have added an additional paragraph as well as orientating the reader towards previous elegant reviews of the area.
Obesity has been associated with earlier onset of T1D and modification of the progression of islet autoimmunity. Given the relationship between AGEs and obesity, adding this piece would strengthen the review.
Thank you for this suggestion. We have added this information to the review in the autoimmunity and AGE sections respectively.
Dietary AGEs:
Please clarify how dry heating baking contributes to AGE formation, and how short heating times lower them.
Thank you for this comment but we disagree that extending this section will enhance this review.
I recommend a figure to explain AGE formation since this is critical to this review.
We have included a very simple figure showing the basic Maillard reaction chemistry (figure 3), however, we believe this is sufficient for the readership of this review as the full chemistry behind AGE formation is so complex that it is not likely to be helpful to the readership of this article. We have however, added references to orientate the reader if they would like to read further about this area.
Kidney disease is not the only complication of diabetes that has been associated with AGEs.
We agree. This has been amended.
Please clarify how high AGEs interfere with ATP synthesis
This has been added.
Islet hormone” should be plural
Thank you. We have corrected it to Islet hormone content.
Overall, there is a lack of clarity on the mechanisms that underlie the involvement of AGE in causing diabetes. It would be helpful to list the mechanisms that have been involved in causing diabetes. Beta-cell dysfunction and beta-cell loss due to autoimmune destruction are usually thought of as two separate mechanisms. Please explain the involvement of AGEs on those two separately.
We believe that this is already evident in the manuscript but have expanded the first area relating to beta cell dysfunction to distinguish it from the latter AGE-RAGE section which describes effects on autoimmunity.
While autoimmune loss of beta-cells is unique to type 1 diabetes, beta-cell dysfunction could happen in T2D as well. Please clarify if the mechanisms above also influence T2D.
Thank you for that suggestion, however we wish this review to focus on Type 1 diabetes, therefore, we have not including substantive discussion regarding Type 2 diabetes. However, we have included a comment at the end of section 2.4 that highlights that many of the mechanisms whereby AGEs influence risk factors for Type 1 diabetes also apply to Type 2 diabetes.
Not all individuals with diabetes have kidney or liver impairment. It would be better “individuals with diabetes who have kidney impairment”. Also, liver impairment is not characteristic of T1D. It appears in T2D in large part due to the association with obesity and NASH and NAFLD. Please clarify
This section has been rewritten and these points clarified.
when added to the presence of autoantibodies”: But autoantibodies are not causal in diabetes. Also, does this imply that AGE could be causing autoantibodies? Please rephrase.
Done.
AGE binding RAGE
The first sentence in the second paragraph seems to belong in the first paragraph. Its meaning is unclear in the current position.
We agree and this has been moved.
“more severe presentation of T1D at diagnosis including DKA and autoantibody status”: autoantibody status is not a sign of severity in T1D. Please clarify.
AA status has been removed from this sentence.
“high sRAGE may be associated with elevated activation of the AGE-RAGE axis”: Could “triggered” be more informative than “associated” ?
We agree and have amended this sentence.
It is unclear why the paragraph “Therapeutically,…” is separated from the previous paragraph that describes interventions to disrupt this axis. Please consider merging those two paragraphs or clarifying why they are separate.
We agree and have merged these paragraphs.
Figure 1:
Legend is all wrong (please see previous text): The description of the stages of preclinical T1D as presented here is highly inaccurate. Stages 1 and 2 by definition do not have symptoms.
This figure legend has been rewritten.
Figure 2:
Legend: Review English language please. Also. Does “Made in Bierender” need to be in the legend?
The English has been reviewed and the made in Biorendor comment removed.
Conclusions
The present conclusion is too vague to be of use. If there a conclusion, please use it to provide a summary, including an overview of the system, list of mechanisms, steps potentially amenable to intervention, etc
We agree and have rewritten the conclusion section.
Reviewer 2 Report
In this review, the authors summarized the interaction between AGEs and RAGE is believed to be a major environmental risk factor and targeting the AGE- RAGE axis may act as a potential therapeutic strategy for T1D prevention. This is an interesting review,I think the authors have done a good job. however, the reviewer has some suggestion before it can be published.
- I think move Figure in the text, which may let readers can better understand.
- Please remove duplicate Figure 1. Line 392.
Author Response
Thank you for your careful consideration of and helpful suggestions for the improvement of this manuscript. Please find out itemised responses below.
- I think move Figure in the text, which may let readers can better understand.
Thank you for this suggestion. We have used the template provided by the journal but we can move the figures to the places where they are cited in the text with their permission.
- Please remove duplicate Figure 1. Line 392.
This has been completed.
Reviewer 3 Report
Chenping Du et al. reviewed the current understanding of the pathogenesis of TID, how advanced glycation end products (AGE) and receptors for advanced glycation end products (RAGE) might be involved and how reducing AGE and RAGE signaling may be crucial for T1D prevention. This review article merits attention in that it is a concise and well-written summary of the relevant issues in the field. Overall, I think this review has the potential to be a very nice resource for those in the field, and the authors do a nice job of walking readers through the recent updates. However, I did have a few comments and concerns about the current version, as detailed below.
-Minor critics:
· It would be good to have a comprehensive schematic representation showing:
o the formation of biological systems AGEs in stages.
o Also, how AGE with RAGE signaling leads to oxidative stress and initiation of the inflammation cascade, diabetic complications, and how reducing AGE and RAGE signaling may be crucial for T1D prevention. This will be easy to grasp for the readers.
· Authors have discussed mainly RAGE and their interactions with AGEs but not scavenger receptors which also exhibit affinity for AGE. Do these receptors have a similar type of interactions and roles as like RAGE in type 1 diabetes? Is it worth discussing this group of receptors?
· Does a treatment like metformin or aspirin affect the AGEs and AGEs/RAGE interactions in type 1 diabetes etc. any recent development?
· Is there any association between RAGE and lipid-lowering drugs in patients with type 1 diabetes?
· How long do AGEs remain in the body circulation, if they remain longer? is it possibly an important diagnostic marker? Their presence may allow monitoring of the progression of the disease and the effectiveness of the therapy.
If the authors could add the below-suggested points that would be kind.
· In general, it’s nice to end an introduction section with a few comments about what the review contains, to help readers understand what’s coming; please consider adding some relevant remarks here.
· The review lacks an overview of the future perspective on the use of antioxidants, like phytochemicals with antioxidants and anti-inflammatory properties to counteract with AGEs/RAGE axis, this should be part of the discussion.
Author Response
Thank you for your careful consideration of and helpful suggestions for the improvement of this manuscript. Please find out itemised responses below.
-Minor critics:
It would be good to have a comprehensive schematic representation showing: the formation of biological systems AGEs in stages.
We agree and have added this as Figure 3.
Also, how AGE with RAGE signalling leads to oxidative stress and initiation of the inflammation cascade, diabetic complications, and how reducing AGE and RAGE signalling may be crucial for T1D prevention. This will be easy to grasp for the readers.
We agree and have modified Figure 4 accordingly.
Authors have discussed mainly RAGE and their interactions with AGEs but not scavenger receptors which also exhibit affinity for AGE. Do these receptors have a similar type of interactions and roles as like RAGE in type 1 diabetes? Is it worth discussing this group of receptors?
We agree that the scavenger receptors such as CD36, SRA/B family and other AGE receptors such as galectin-3 (so called AGE-R3) are important although little is known about them in the pathogenesis of Type 1 diabetes. Hence, we have added a paragraph on these receptor/binding proteins and their associations with complications or processes related to T1D pathogenesis.
Does a treatment like metformin or aspirin affect the AGEs and AGEs/RAGE interactions in type 1 diabetes etc. any recent development?
Obviously, most studies using these agents focus on Type 2 diabetes. There have been some preclinical studies using salicylates at high doses (https://doi.org/10.1371/journal.pone.0078050 ) but there are some contraindications for these agents in people with Type 1 diabetes. For metformin, studies have almost exclusively looked at its role in reducing diabetes complications in Type 1 diabetes, and it is commonly prescribed to these individuals if they develop issues with insulin sensitivity from their exogenous insulin therapy or have difficulty in controlling their blood glucose concentrations. There is one older article showing that metformin has no effect on type 1 diabetes development in the preclinical NOD mouse model (PMID: 9230345). However, as we are focused specifically on the role of AGEs and the AGE-RAGE axis in Type 1 diabetes development we feel that discussing this is outside of the scope of this review.
Is there any association between RAGE and lipid-lowering drugs in patients with type 1 diabetes?
Both statins (PMID: 2396666) and fenofibrate (https://doi.org/10.1186/s12944-020-01397-2) have been shown to influence the circulating concentrations of the decoy competitive receptor soluble RAGE in individuals with long standing Type 1 diabetes but not in Type 1 diabetes development. As this review is focussed on type 1 diabetes development we feel it is outside of the scope of the review to discuss lipid lowering therapies in this context.
How long do AGEs remain in the body circulation, if they remain longer? is it possibly an important diagnostic marker? Their presence may allow monitoring of the progression of the disease and the effectiveness of the therapy.
AGE concentrations in the circulation are already used as diagnostic markers – both HbA1C and fructosamine albumin are used in routine clinical practice globally. Please see line 299 of the marked up version where we already allude to this. We have additionally added more detail regarding the measurement of AGEs in the skin which has been shown to be a good biomarker of long-term AGE production and micro and macrovascular complications of diabetes please refer to page 8 of the marked up review.
If the authors could add the below-suggested points that would be kind.
In general, it’s nice to end an introduction section with a few comments about what the review contains, to help readers understand what’s coming; please consider adding some relevant remarks here.
Thankyou for this suggestion, we have added some extra detail here to expand upon this, however, we were reluctant to expand this section beyond a single sentence.
The review lacks an overview of the future perspective on the use of antioxidants, like phytochemicals with antioxidants and anti-inflammatory properties to counteract with AGEs/RAGE axis, this should be part of the discussion.
Thank you for this suggestion. However, we have chosen not to include this in this review which is already exceeding the word length required by the editorial board.